# Synergy between land use and climate change increases future fire risk in Amazon forests

Yannick Le Page[1], Douglas Morton[2], Corinne Hartin[3], Ben Bond-Lamberty[3], José Miguel Cardoso Pereira[1],

George Hurtt[4], Ghassem Asrar[3]

[1] Centro de Estudos Florestais, Instituto Superior de Agronomia, Universidade de Lisboa, Tapada da Ajuda, 1349-017 Lisbon, Portugal
[2] NASA Goddard Space Flight Center, Greenbelt, MD 20771, USA
[3] Pacific Northwest National Laboratory, Joint Global Change Research Institute, University of Maryland, College Park, MD 20740
[4] Department of Geographical Sciences, University of Maryland, College Park, MD 20740

*Correspondence to*: Yannick Le Page (niquya@gmail.com)

**Abstract.** Tropical forests have been a permanent feature of the Amazon basin for at least 55 million years, yet climate change and land use threaten the forest's future over the next century. Understory forest fires, common under current climate in frontier forests, may accelerate Amazon forest losses from climate-driven dieback and deforestation. Far from land use frontiers, scarce fire ignitions and high moisture levels preclude significant burning, yet projected climate and land use changes may increase fire activity in these remote regions. Here, we used a fire model specifically parameterized for Amazon understory fires to examine the interactions between anthropogenic activities and climate under current and projected conditions. In a scenario of low mitigation efforts with substantial land use expansion and climate change – the representative concentration pathway (RCP) 8.5 – projected understory fires increase in frequency and duration, burning 4-28 times more forest in 2080-2100 than during 1990-2010. In contrast, active climate mitigation and land use contraction in RCP4.5 constrain the projected increase in fire activity to 0.9-5.4 times contemporary burned area. Importantly, if climate mitigation is not successful, land use contraction alone is very effective under low to moderate climate change, but does little to reduce fire activity under the most severe climate projections. These results underscore the potential for a fire-driven transformation of Amazon forests if recent regional policies for forest conservation are not paired with global efforts to mitigate climate change.

## 1    Introduction

Tropical forests face an unprecedented suite of environmental changes from regional and global human activities. Regional activities include forest conversion for agricultural land uses (Hansen et al., 2013), forest degradation from logging and fire (Asner et al., 2005; Morton et al., 2013), and fragmentation (Laurance and Williamson, 2001). Climate change is primarily

driven by extra-regional activities, as rising greenhouse gas emissions from energy production and transport mostly originate outside of tropical forest regions (Quéré et al., 2016). Synergies between direct human activities and climate change may accelerate the transformation of tropical forests, based on positive feedbacks among ecosystem productivity, regional climate, species composition and forest disturbances (Brienen et al., 2015; Davidson et al., 2012; Malhi et al., 2009). Forest

fires are one potential mechanism for a disturbance-driven dieback of Amazon forests (Balch et al., 2015; Barlow et al., 2016; Cochrane et al., 1999; Longo et al., n.d.).

Fires during the Amazon dry season result from human ignitions for large-scale deforestation (Morton et al., 2008), shifting cultivation (Thrupp et al., 1997) and agricultural management (Cochrane, 2002). Intentional and accidental fires can escape

into adjacent forests and slowly spread through leaf litter and downed woody debris on the forest floor. Often, understory forest fires extinguish at night, when relative humidity increases. During extended droughts, however, or other periods with lower night time humidity, understory fires may spread for days or weeks (Cochrane et al., 1999; Morton et al., 2013). Because the expansion of the fire perimeter over multiple days generates quadratic growth in burned area, these long duration fires are an essential aspect of fire ecology in the Amazon. Most plant species are poorly adapted to survive even

low-intensity fires (Balch et al., 2015; Barlow and Peres, 2008; Malhi et al., 2008). The resulting tree mortality and canopy openings alter the forest microclimate and favour invasion by grass species; combined, changing fuels and microclimate in burned forests can create a positive fire feedback (Balch et al., 2008; Barlow and Peres, 2008; Cochrane et al., 1999; Silvério et al., 2013).

Climate and land use projections for the Amazon region may increase the regional extent and frequency of forest degradation from fire. Agricultural activities are projected to expand into the central and western Amazon (Davidson et al., 2012; Rosa et al., 2013; Soares-Filho et al., 2006), regions where humid conditions currently limit the risk of escaped fires (Le Page et al., 2010). During drought years or with climate change, however, the influence of human ignitions on fire activity could greatly expand. Satellite observations over the last 40 years have confirmed the importance of these fire-climate-land use

interactions (Alencar et al., 2015; Chen et al., 2013; Laurance, 1998), as did paleorecords of charcoal accumulation rates inferred from sedimentary cores (Bush et al., 2007; Cordeiro et al., 2014). Projections of Amazon fire activity also suggest strong synergies between climate change and anthropogenic expansion scenarios (Cardoso et al., 2003; Le Page et al., 2010; Silvestrini et al., 2011), but previous work focuses primarily on deforestation and agricultural burning. These types of fires are managed, burn different types of fuel, and are generally of short duration, thus provide few insights about the ecology of

slow-moving, multi-day understory fires. Understory fires are difficult to detect using satellite data because they do not burn the forest canopy, and only a few studies have inferred their extent in small regions to explore their dynamics and drivers (Alencar et al., 2006, 2004; Ray et al., 2005). However, a method was recently developed to detect understory forest fires using multi-year satellite image time series (Morton et al., 2011a, 2013). These Amazon-wide observations provide a critical foundation to develop simulations of understory fire dynamics under future climate and land use scenarios.

Here, we investigated the combined influence of future land use and climate change on Amazon understory fires. Our analysis used a regional fire model that was specifically adapted to simulate understory fires and parameterized using satellite-based estimates of understory fire activity (see Methods). We compared modelled and observed understory fires to characterize the climate-fire interactions that promote increased burned area and larger fires in drought years. Using the calibrated model, we then examined fire projections under land use and climate change scenarios from the Representative Concentration Pathway 4.5 (RCP4.5 (Thomson et al., 2011)) – with substantial mitigation efforts, and the RCP8.5 (Riahi et al., 2011) – essentially unmitigated (Figure 1). Fire projections under these contrasting scenarios evaluate potential synergies between human land use and climate change for a fire-driven transformation of Amazon forests.

## 2 Methods

### 2.1 The HESFIRE model

HESFIRE is a fire model of intermediate complexity seeking to combine the explicit fire modeling of dynamic vegetation-fire models (DGVM-fire models) with the performance of statistical fire models, while addressing some of their issues (Le Page et al., 2015). In particular, land cover distribution in HESFIRE is inferred from contemporary observations, avoiding error propagation from the vegetation scheme to the fire module, which is a recurrent challenge in DGVM-fire models (Kelley et al., 2013; Kelley and Harrison, 2014; Wu et al., 2015). HESFIRE was also designed to represent multi-day fires, tracking each individual fire on a 12-hour timesteps, whereas other global fire models have a maximum fire duration of 1 day (Arora and Boer, 2005; Li et al., 2012; Thonicke et al., 2010). The model has been applied at global scale (Le Page et al., 2015), and in a sensitivity experiment to reveal the propagation of uncertainties in land cover and climate input data to the fire activity outputs (Le Page, 2016). HESFIRE was designed to facilitate the development of regional versions in studies such as this one, with the integration of an innovative data assimilation component to regionally adjust the parameterization of fire-drivers based on observed fire dynamics.

HESFIRE is organized in three modules, with specific drivers for fire ignition, spread, and termination (Figure S1a, (Le Page et al., 2015)). Each module's parameterization is derived from literature values and through observation-data assimilation methods:

- *Fire ignitions*. The frequency of human ignitions increases as a function of grid-cell level land use density (cropland + urban areas) but decreases as a function of the national Gross Domestic Product (GDP). Natural ignitions are a function of cloud-to-ground lightning strikes, estimated from convective precipitation, and a probability of ignition per strike.

- *Fire spread*. Fire spread rates vary as a function of weather conditions (relative humidity, temperature, wind speed), soil moisture (a proxy for fuel moisture), and fuel structure (forest, shrub and grass).

- *Fire termination*. Four factors control the termination of fires: a) a change to non fire-prone weather conditions (e.g. fires terminate when relative humidity increases above 80%); b) low fuel availability (the probability of termination is higher in sparsely-vegetated landscapes); c) landscape fragmentation (the fraction of a grid-cell covered by croplands, urban areas, water bodies, bare areas, and burned areas over the last 8 months); and d) fire suppression efforts, which intensify with higher land use density and GDP, but become less efficient under increasingly fire-prone weather.

## 2.2 Adjustments to the original HESFIRE model

Two changes from the original HESFIRE model were implemented for regional simulations of Amazon understory fires. First, land use and land cover data from Globcover (Bontemps et al., 2011) were replaced by MODIS (Friedl et al., 2010). Although there is no comparison study of both datasets in the Amazon, MODIS patterns appeared more consistent with the contemporary distribution of land use, as inferred from expert knowledge in the team and from a comparison with other sources of information (e.g. (Soares-Filho et al., 2014)). Second, the representation of fires as plain ellipses was revised, as proposed in the global HESFIRE evaluation (Le Page et al., 2015) and further supported by the uneven and patchy contours of Amazonian understory fires (Figure S7). A new equation was developed in this study to compute the area that actually burns as a fraction of the plain ellipse, driven by landscape fragmentation and fire weather:

$$BA = E \times \left(1 - F_n^{F_{exp}}\right) \times \left(1 - RH_n^{RH_{exp}}\right) \times \left(1 - SW_n^{SW_{exp}}\right) \times \left(1 - T_n^{T_{exp}}\right) \quad (2)$$

where $E$ is the full ellipse area, $F_n$, $RH_n$, $SW_n$ and $T_n$ the values of fragmentation, relative humidity, soil moisture and temperature normalized within their range of influence, and $F_{exp}$, $RH_{exp}$, $SW_{exp}$ and $T_{exp}$ the optimized shape parameters controlling their specific fire-driving relationship (Le Page et al., 2015).

## 2.3 Parameterization of the regional HESFIRE version

Parameter optimization was performed using a Markov Chain Monte Carlo approach as described in the original HESFIRE paper (Le Page et al., 2015). A selection of 6 parameters key to fire regimes in the Amazon were optimized: the influence of relative humidity (RH) and soil moisture (SW) on fire spread (one parameter each), the frequency of human ignitions as a function of land use area (two parameters), the fire suppression effort as a function of land use area (one parameter), and the influence of landscape fragmentation on fire termination (one parameter). The optimization was performed using about 20% of the grid-cells in a satellite-derived understory fires dataset over the 1999-2010 period (see below). The optimization metric combines average burned area and inter-annual variability at the grid-cell level:

$$Opt_{index} = \frac{\sum_{gc=1}^{n}(M_f - O_f)^2 + \sum_{gc=1}^{n}\left(1 - IAV_{corr}(M_f, O_f)\right)}{n} \qquad (1)$$

where *Mf* and *Of* are modeled and observed burned areas for the grid-cell *gc*, and *IAVcorr(Mf,Of)* the correlation between observed and modeled inter-annual variability. Note that the use of a grid-cell level metric means that fire patterns and dynamics aggregated at the regional scale are not directly optimized.

## 2.4   Observation-derived fire data

The regional optimization of HESFIRE and its evaluation were performed using forest fire data derived from time series of MODIS data and the Burned Damage and Recovery (BDR) algorithm (Morton et al., 2011a, 2013). The BDR approach detects the spectral trajectory of canopy damage from understory fires and recovery in subsequent years. The multi-year method discriminates understory fires from other disturbances such as deforestation or logging (Morton et al., 2011a).

## 2.5   Land use and climate scenarios

Future fire projections considered two contrasting scenarios of climate and land use in 2080-2100 from the Inter-governmental Panel on Climate Change Assessment Report 5 (IPCC AR5). The Representative Concentration Pathway 4.5 (RCP4.5 (Thomson et al., 2011)) represents a world where society deploys substantial decarbonization and ecosystem conservation measures to limit the increase in greenhouse gas radiative forcing to +4.5W.m-2 in 2100. In the Amazon basin, these measures halt the current trend of land use expansion and support some reforestation (Figure 1). The RCP8.5 (Riahi et al., 2011) limits radiative forcing to +8.5W.m-2, which poses very few development constraints and leads to continued agricultural expansion in the Amazon (+216% land use) as well as substantial climate change (Figure 1).

For both scenarios, HESFIRE was run with future land-use distributions from the land harmonization processing developed for the RCPs (Hurtt et al., 2011) and climate change projections from 8 of the 20+ climate models contributing to the IPCC report, covering a wide range of climate sensitivity in the Amazon (Figure 1). Climate models were selected to capture a broad range of projected climate sensitivity in the Amazon (Figure 1). For each climate variable, monthly absolute changes from 1990-2010 to 2080-2100 were computed using the RCMIP5 package (https://github.com/JGCRI/RCMIP5). These changes were then applied to the 1990-2010 climate data used in HESFIRE (Le Page et al., 2015), with each bi-daily data for a given month being altered by the same monthly change.

Changes in fire practices and fire suppression other than those related to land use expansion and contraction were not considered. Such changes will depend on the co-evolution of many factors, including technological development (e.g. alternatives to fire use), rural versus urban environments, and economic changes. HESFIRE and most global fire models use GDP as a proxy of those drivers (e.g. widespread fire use in Africa), but it is unclear whether this relationship will hold in

the future (e.g. will increasing GDP in South America lead to less fire use?). Consequently, our analysis assumed a continuation of current fire practices by using today's GDP for future fire projections.

## 3    Results

### 3.1    Model evaluation

The regional version of HESFIRE reproduced the observed spatial patterns of average fire activity, including the clear boundary between fire-affected forests along the arc of deforestation and mostly fire-free forests in more humid regions of the central and western Amazon with less agricultural activity (Figure 2a, ,Figure S2, see also reply to referee comment #1 in discussion paper). Importantly, HESFIRE captured the interannual variability in Amazon fire dynamics, such as the impacts of the 2005 and 2007 droughts (Figure 2b, Figure S4), and the modeled fire size distribution was consistent with
observations (Figure 2c). Larger fires were more common in high-fire years (e.g. 2005, 2007, and 2010), while smaller fires contributed a greater proportion of total burned area in low fire years (e.g. 2000, 2003, and 2009). The main discrepancies between satellite observations and HESFIRE results were an overestimation of basin-wide burned area in 1999, as climate data indicated more widespread drought conditions than areas with large-scale fire damages (Figure S4), and an underestimation of the fire hotspot in the Brazilian state of Maranhão in the eastern Amazon. Overall, the model formulation
in line with key understory fire processes (e.g. multiday fires) and its performance on a wide range of fire metrics provide a strong basis for evaluating future fire regimes in the Amazon.

### 3.2    Projected changes in understory fires

HESFIRE simulations projected much higher fire activity at the end of the 21$^{st}$ Century under scenarios combining drier climate conditions and agricultural expansion (Figure 3). In the essentially unmitigated RCP8.5, the projected annual burned
area in 2080-2100 was 850% that of today (50$^{th}$ percentile among the 8 climate model replicates; 10$^{th}$ and 90$^{th}$ percentiles: 400% and 2800%). In this scenario, extreme fire years in the observational record (e.g., 2005, 2007 and 2010) would be below-average fire years by the end of the century. Model projections under this scenario suggest a 48% reduction in the area of the Amazon basin that would remain "fire-free" (with average annual burned area <0.05%), decreasing from 67% today to 35% under the RCP8.5 conditions (10$^{th}$ to 90$^{th}$ percentile model range: 11% to 49%).


In the RCP4.5 scenario, global climate mitigation policies and land use contraction limited increases in fire activity to levels well below those in most RCP8.5 projections (Figure 3). Annual burned area was 165% that of today for the 50$^{th}$ percentile, ranging from a slight reduction (87%) for the 10$^{th}$ percentile to 540% for the 90$^{th}$ percentile spread among models. The RCP4.5 scenario also limited the encroachment of fire activity into interior Amazon forests; the area of fire-free forests
decreased by 16% only,  covering 56% of the study area by the end of the century (10$^{th}$ to 90$^{th}$ percentile range: 28% to 69%).

Higher future fire activity results from an increase in the number and size of fires in both scenarios (Figure 4). The higher number of fires is related to anthropogenic ignitions, more frequent with widespread land use expansion in RCP8.5, amplified by a longer period of fire-prone conditions. Despite a contraction of land use in the RCP4.5 and associated

reduction in anthropogenic ignitions, projected climate changes lead to more numerous understory fires based on a higher probability of escape and longer dry seasons over the Amazon basin (Aragão et al., 2007). The shift in the size distribution towards larger understory fires reflects longer periods of consecutive dry days in future decades - including low nighttime relative humidity – conditions that allow the spread of slow-burning fires over larger areas. The longest modeled fires last 10-15 days under current conditions, but fires burn for as long as 25-30 days over most of the Amazon basin under the

RCP8.5 scenario by the end of the 21$^{st}$ century (Figure 4).

### 3.3    Sensitivity of fire projections to climate and land use

A sensitivity analysis of future fire activity suggests that climate change is the most important driver of increased Amazon understory fire activity by 2080. We explored the roles of land use change, climate change, and their interactions on fire projections based on a factorial experiment whereby changing conditions are restricted to subsets of the model variables

(Figure 5, Figure S5). Holding current land use activity constant, climate change alone would be sufficient to generate large increases in both total burned area and the share of the Amazon basin affected by understory fires, contributing 48% and 75% of their changes in the full RCP8.5 scenario, respectively (Figure 5a). Conversely, the expansion of land use activities under current climate regimes would contribute only 18% and 47% of those changes (Figure 5b). Climate change thus emerges as the dominant concern for future increases in fire activity, with a clear potential to expand understory fires into

interior Amazon forests, as stronger and longer dry seasons elevate fire risk beyond the current arc of deforestation. Land use activities play a key role in modulating fire outcomes within the climate-driven boundaries of flammable forests; under a given climate, land use contraction can achieve substantial fire mitigation, while large scale land use expansion amplifies the projected increase in understory fires from synergies between human ignitions and climate-driven fire risk (Figure 5c,d,e).

Importantly, the level of interaction between climate and land use scenarios depends on the magnitude of climate change projected by the 8 climate models used in the analysis (Figure 1b), with important implications for mitigation strategies (Figure S5). Land use contraction in RCP4.5 limited fire activity under low to moderate climate change, but was only marginally effective under a large-scale drying of the Amazon projected by some models. For example, with unmitigated climate change (RCP8.5), land use contraction in the RCP4.5 scenario avoided 93% of the full RCP8.5 fire increase under

the conservative IPSL climate projections, but only 8% under the much drier ACCESS climate projections (Figure S5r). In the latter case, the interior eastern Amazon becomes an area of essentially unfragmented fire-prone forest, where residual land use activity remains a source of ignitions, and escaped fires, albeit less frequent, can reach larger sizes (Figure S6b). In the opposite situation, where climate mitigation is successful but land use expansion continues, fire size is clearly

constrained (Figure S6c) leading to a smaller increase in total burned area (Figure S5s), with 77% of the ACCESS RCP8.5 fire increase avoided, for example.

## 4 Discussion

Scenarios of future fire activity underscore the importance of climate change mitigation to prevent the expansion of damaging understory fires in the Amazon. Climate change scenarios that maintain sufficient moisture to insulate interior Amazon forests from understory fires can avoid disturbance-driven forest losses even when anthropogenic ignitions are abundant (Figure 5b). Some climate change is likely unavoidable, however, raising the importance of climate modeling efforts to anticipate changing fire regimes. The current generation of climate models are better able to reproduce contemporary moisture regimes in the Amazon, but there is little consensus among future projections (Duffy et al., 2015; Joetzjer et al., 2013; Yin et al., 2013). The emerging picture from state-of-the-art models suggests longer and stronger dry seasons, with some evidence that pessimistic models are more realistic (Boisier et al., 2015). The evolution of the El Nino Southern Oscillation (ENSO) and the Atlantic Multidecadal Oscillation (AMO) dynamics in a warmer world will be critical for the future of Amazon forests, given the widespread influence of these climate modes on drought conditions in the region (Chen et al., 2011, 2016). Warm phases of ENSO and AMO typically trigger basin-wide droughts conducive to large fire events (Chen et al., 2011). The synergy between their periodicity, magnitude and other climate change impacts will be a major driver of Amazon fire activity in coming decades.

Model results with RCP4.5 land use projections suggest that a significant and regional-scale reduction in agricultural activities and landscape fragmentation can disrupt these climate-fire interactions. This is consistent with contemporary studies (Alencar et al., 2004; Morton et al., 2013), and with the clear signature of land use activities in charcoal paleorecords since the onset of Amazonian agriculture (Bush et al., 2007; Cordeiro et al., 2014). In fact, (Bush et al., 2017) analysed a 6900 years sedimentary record in western Amazonia, and found that the ENSO-fire signal was strongly expressed at times of widespread agricultural activity, being otherwise undetectable in the record and most likely absorbed by natural vegetation. Deforestation rates have largely declined over the last decade (-70% in Brazil, (Nepstad et al., 2014)), and several Amazon countries have committed to ambitious reforestation targets through their Intended Nationally Determined Contributions (INDC) to the United Nations Framework Convention on Climate Change (e.g. 12 million hectares by 2030 in Brazil (Federative Republic of Brazil, 2015)). The RCP4.5 land use scenario is thus fairly consistent with these recent regional developments, highlighting the potential for environmental policies, enforcement, and satellite-based monitoring to alter the trajectory of agricultural expansion and buffer climate change impacts.

However, recent droughts and model projections confirm widespread vulnerability to forest degradation from fire remains. Declining deforestation in the Brazilian Amazon since 2004 had little impact on understory fires, with peak fire damages in

2005, 2007 and 2010 (Chen et al., 2013; Morton et al., 2013). HESFIRE simulations are consistent with observational data, confirming that land-use policies have limited effectiveness under anomalously dry conditions. The hybrid scenario combining RCP4.5 land use with RCP8.5 climate change is particularly relevant in this context (Figure S5r). This combination depicts the disconnect between recent regional achievements on forest conservation and remaining global challenges to achieve substantial reductions in global greenhouse gas emissions. Climate drivers of contemporary fire events are well-understood and anticipated by seasonal forecasts ((Chen et al., 2011), see https://www.ess.uci.edu/~amazonfirerisk/) yet recent fire emergencies overwhelmed existing prevention and suppression capabilities. Regular, basin-scale droughts in some of the hybrid RCP4.5/RCP8.5 scenario projections would thus represent formidable challenges for regional fire management and forest resource conservation.

Anticipating these outcomes is key to connect our understanding of fires in the Amazon to decision-makers. Challenges remain, but data and knowledge assimilation in tools such as HESFIRE can provide support to evaluate the coevolution of agricultural and conservation objectives in alternative policy scenarios. The main assets of our analysis are the explicit representation of multi-day fires, showing the potential for longer-duration fires, and the assimilation of satellite-derived data that discriminate understory fires from other types of burning. Results from the factorial experiment also suggest that the sensitivity of fire activity to individual natural and anthropogenic variables is consistent with our knowledge of fire dynamics in the Amazon (Figure S5). A number of ecological processes are not considered in HESFIRE, however, most notably the lack of a dynamic vegetation scheme to account for climate-vegetation-fire feedback. Higher tree mortality and decreased evapotranspiration induced by fires and drought stress alter forest structure and facilitate the invasion of fire-prone grasses, and would thus likely enhance the prospect of a fire-driven transformation of the Amazon under the RCP8.5 climate projections (Brando et al., 2014; Malhi et al., 2008). Representing these interactions is challenging because they involve many processes from local- to global- scale, including plant species resilience, fuel dynamics, invasive species dynamics, landscape fragmentation, climate dynamics and the water cycle. Efforts to improve the sophistication of regional fire models remains an important goal for future research.

## 5    Conclusion

Recently, regional land use policies have been successfully implemented in the Amazon basin, but the outcome of international negotiations on global climate change is highly uncertain. This paper highlights how the fire challenge is likely to grow if global climate mitigation is not successful. Infrequent fires are not disastrous, but regular, basin-scale fire events would be unmanageable and accelerate forest degradation. Additionally, climate projections leading to increased fire duration and size (relative humidity, temperature, and fuel moisture) are also conducive to higher fire intensity, leading to greater tree mortality (Balch et al., 2011; Brando et al., 2014) and hindering suppression efforts. Further concerted local action may thus be necessary to consider forest degradation along with deforestation and reforestation objectives.

Intensifying existing agricultural areas and switching to fire-free land management could substantially reduce the extent of frontier forests exposed to understory fires. The large wealth of comparative data provided by diverse regional land use policies, various land management regimes (e.g. protected areas) and a range of large-scale droughts provide a unique opportunity to infer efficient fire management strategies (Nepstad et al., 2006, 2014; Nolte et al., 2013; Soares-Filho et al., 2010). Overall, however, Amazon fires are a global climate mitigation problem, or a very difficult one to solve locally.

(1)

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

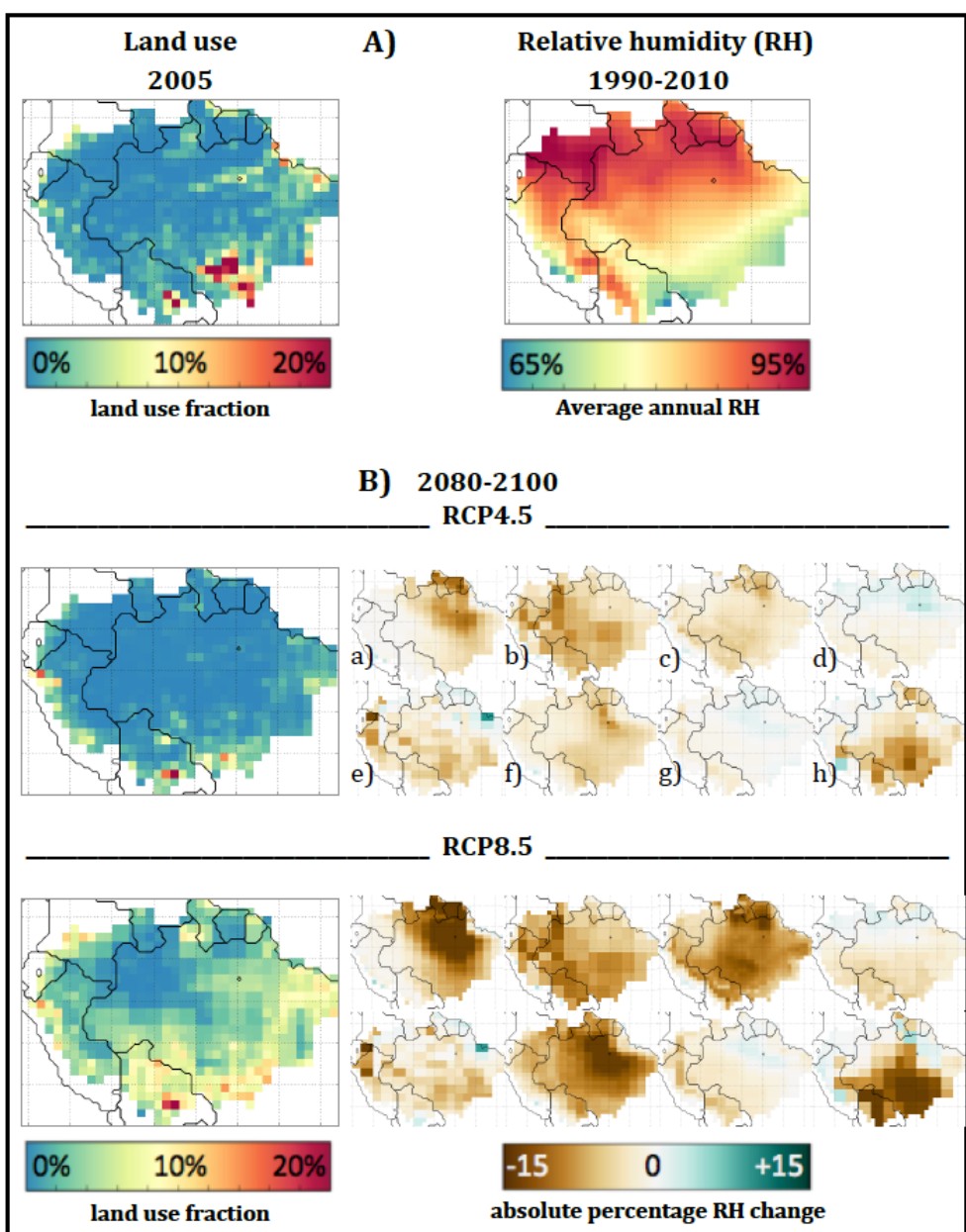

**Figure 1: Current and projected climate and land use in the Amazon Basin. A)** Land use fraction in 2005 (croplands + urban lands) from the MODIS (Friedl et al., 2010) and average annual relative humidity over 1990-2005 from NCEP (Kanamitsu et al., 2002). **B)** Land use fraction in 2080-2100 in the RCP4.5 and RCP8.5 scenarios (Hurtt et al., 2011) and change in average relative humidity from 8 of the IPCC AR5 climate models: a) ACCESS1-3; b) 3CanESM2; c) GFDL-ESM2M; d) IPSL-CM5A-MR; e) HadGEM2-CC; f) MIROC-ESM; g) GISS-E2-H; h) CESM1-CAM5.

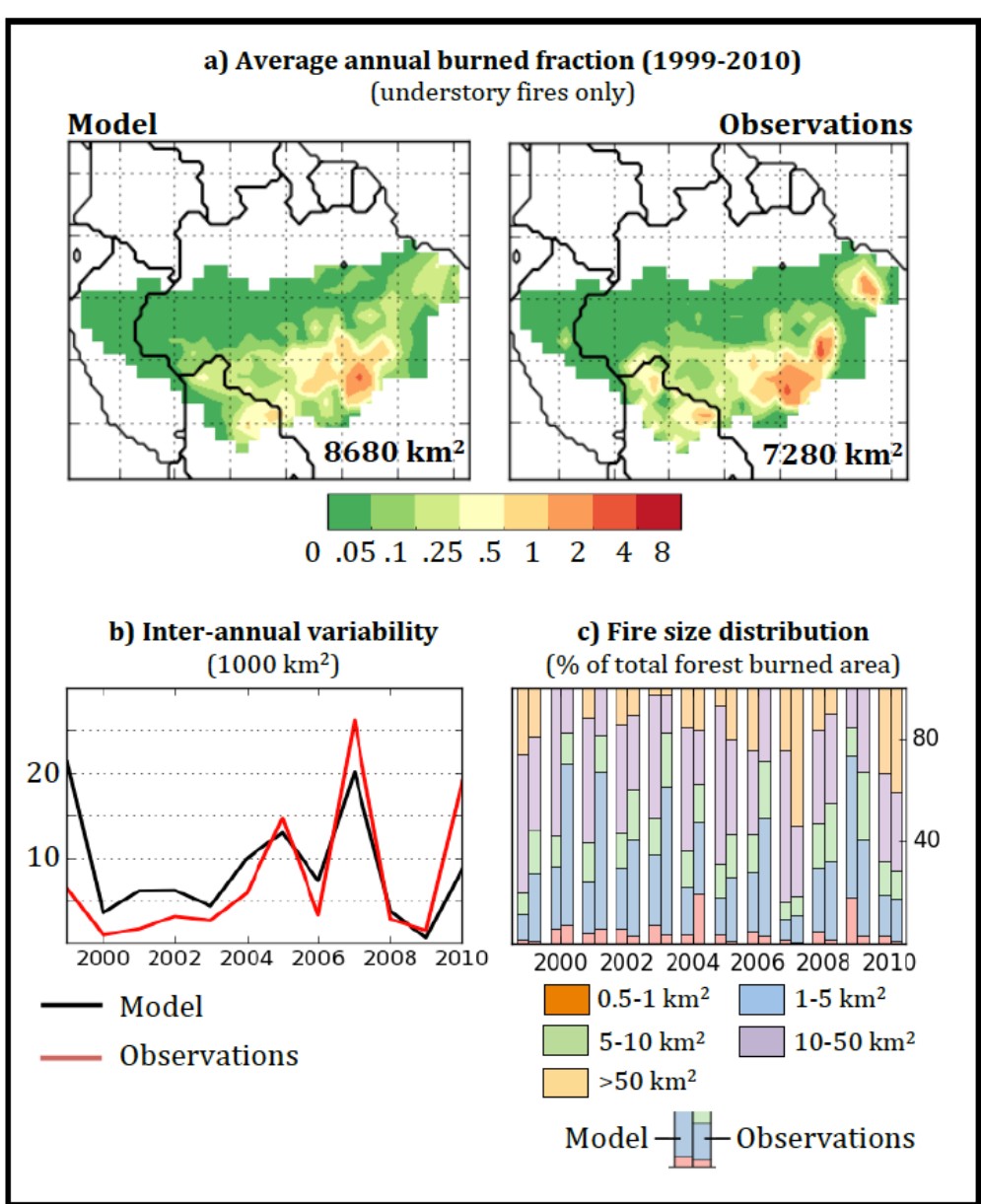

**Figure 2: Spatial and temporal patterns of modeled and observed understory fires in the southern Amazon. Observation data are specific to understory fires (Morton et al., 2011b), i.e. do not include deforestation or agricultural fires.**

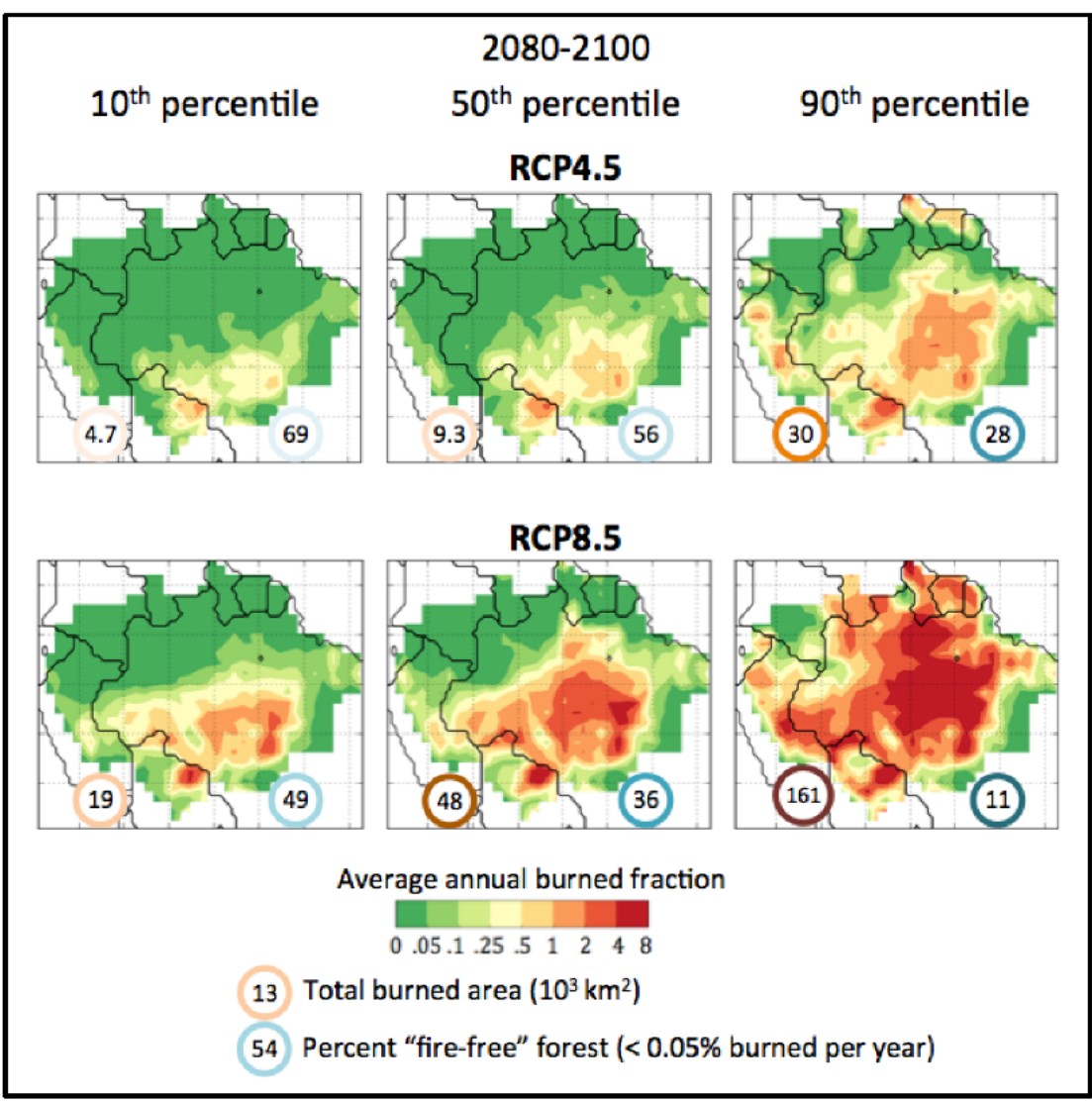

**Figure 3: Annual burned fraction projected in HESFIRE for 2080-2100 varies across models and climate scenarios. The 10th, 50th and 90th percentile values indicate variability from 8 climate models (see Methods).**

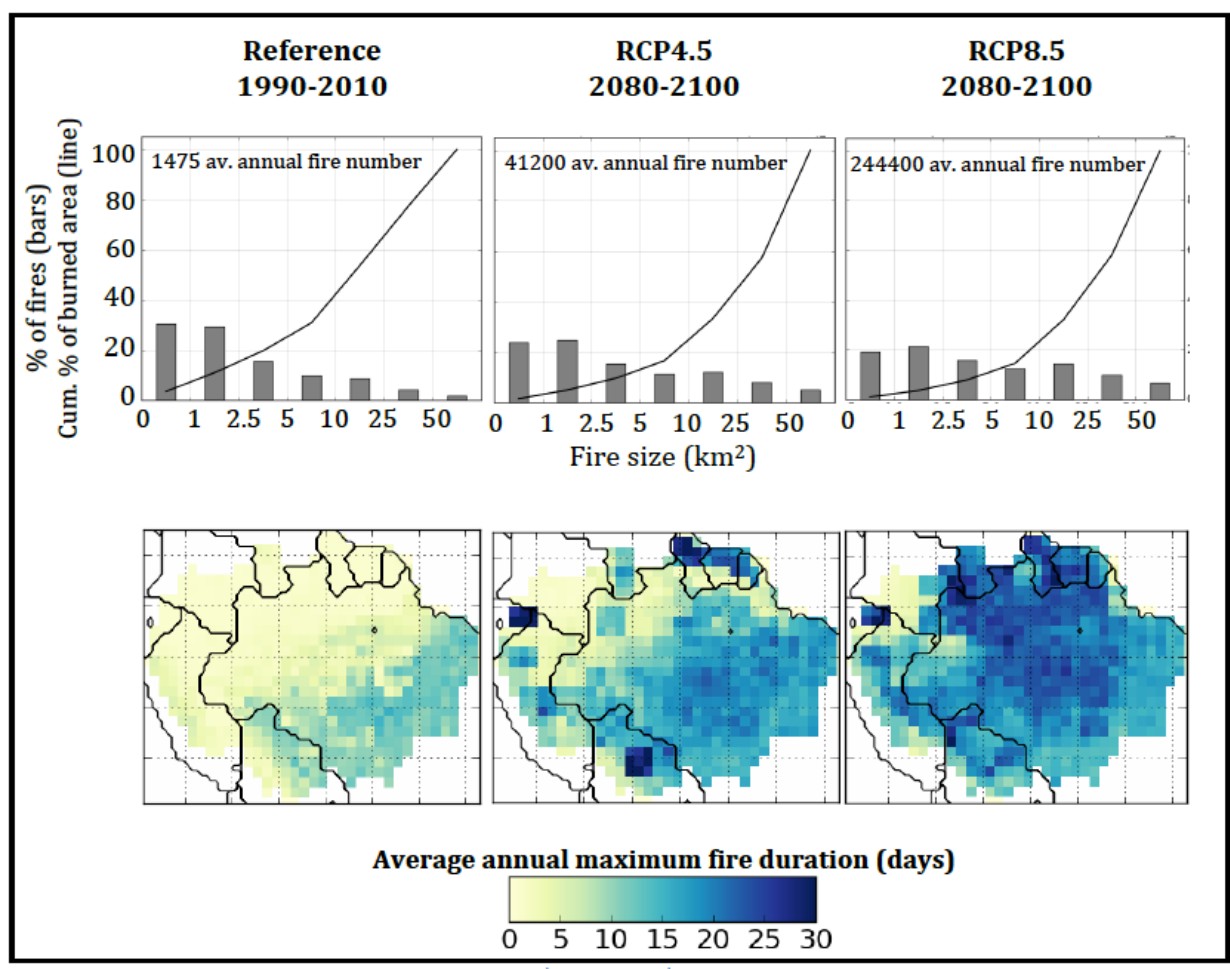

**Figure 4: Mean fire size increases under both RCP4.5 and RCP8.5 climate scenarios (top) based on climate conditions that permit longer fire duration by the end of the 21st century (bottom).**

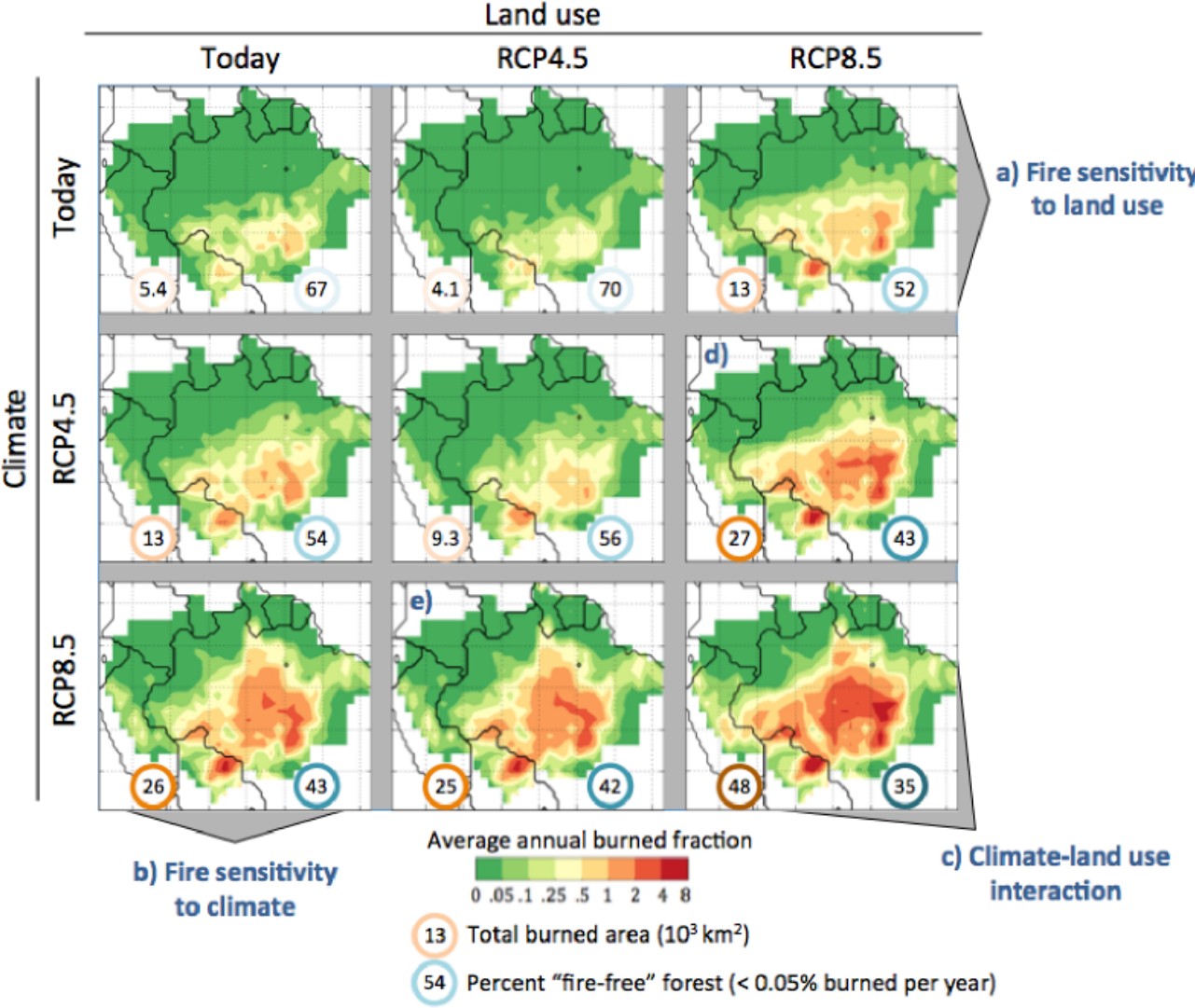

**Figure 5: Sensitivity of understory fire activity to land use and climate change. These results are obtained from a factorial experiment running HESFIRE with a) land use change only, b) climate change only; and c) both land use and climate change. d) and e) are scenarios combining land use from one RCP and climate change from the other. Fire maps are the 50th percentil (median) at the grid-cell level (as in Figure 3).**

In supplementary material:

Figure S1,Figure S2,Figure S3,Figure S4,Figure S5,Figure S6,Figure S7