# Peer review of "Synergy between land use and climate change increases future fire risk in Amazon forests"

_Earth System Dynamics, 2017_

## Referee Comment (RC1) · W.D. Gosling (Referee) · 21 Jul 2017

As a specialist in past ecological change, and having worked with reconstructing fire histories in the Amazon basin, I was interested to read the discussion manuscript by Le Page et al. related to the likely impact of future climate change on fire in Amazonia. In this manuscript Le Page et al. develop the global intermediate complexity Human-Earth System FIRE (HESFIRE) model (Le Page et al. 2015) to explore fire dynamics in the Amazon basin. The new outputs presented here project how understory fire activity could change within Amazonia over the next c. 100 years under two different future climate change scenarios (Representative Concentration Pathways 4.5 and 8.5; Thompson et al. 2011, Riahi et al. 2011). The HESFIRE model is evaluated favourably against observed fire activity between 1999 and 2010; interestingly the new parame-

terisation of the model (this manuscript) is shown to be a considerable improvement on the original parameterisation used in the global model (compare Figure 2 with Figure S2). The authors predict that over the coming decades climatic change will lead to changes in relative humidity, temperature and fuel moisture that will lead to increased duration and size of fires, and that fires will impact a larger area of the basin.

The manuscript presented by Le Page et al. is clearly written, well organised and contains clear figures (notwithstanding minor comments below). The model parameters make intuitive sense to me but I am not an expert in this area. The conclusion that the synergy of land-use (humans) and climate change are likely to increasing fire risk in Amazon basin is reasonable based on the data presented. It also does not come as a surprise to me as there is a long history of human-climate-fire interactions within the Amazon basin. Fire (most probably triggered by early human populations) has been a component of Amazonia for c. 8,000-9,000 years (e.g. Bush et al. 2007; Hammond et al. 2007), and the impact of this fire has likely been modulated by past climate change (Bush et al. 2008; Bush et al. 2017; Cordeiro et al. 2014). Indeed Bush et al. (2017) make a specific link past changes in the El Nino Southern Oscillation (ENSO), human activity, and changes in the Amazonian fire regime (based on fossil charcoal records). Given that the message from the past, and the projections for the future, seem to be providing consistent message (humans and climate synergistically modifying fire regimes within Amazonia) it might be interesting to add a short section of discussion to this manuscript to make this comparison. This could fit well at the start of the discussion section where ENSO is discussed in the context of future predictions.

MINOR COMMENTS

Page 3, line 10-20: There seems to be a bit of a jump in the HESFIRE model. It would be useful to add some linking text that makes clear what this model is, how it has been previously used and why the work presented her is a good next step.

Page 4, line 13: Additional space between "Figure" (also: page 6 line 11, page 7 line

30, page 8 line 29, page 10 line 10).

Page 8, line 33: Referencing anonymous new data seems a bit odd. Can you find a clearer way of referencing the source of these data?

Figures

Various fonts are used on the different figures. I think it would help the manuscript to look more coherent if these were standardised.

ADDITIONAL SUGGESTED REFERENCES

Bush, M.B., Correa-Metrio, A., van Woesik, R., Shadik, C.R. & McMichael, C.N.H. (2017) Human disturbance amplifies Amazonian El Niño–Southern Oscillation signal. Global Change Biology 23, 3181-3192. http://dx.doi.org/10.1111/gcb.13608

Bush, M.B., Silman, M.R., McMichael, C. & Saatchi, S. (2008) Fire, climate change and biodiversity in Amazonia: A Late-Holocene perspective. Philosophical Transactions of the Royal Society B: Biological Sciences 363, 1795-1802. http://dx.doi.org/10.1098/rstb.2007.0014

Bush, M.B., Silman, M.R., de Toledo, M.B., Listopad, C., Gosling, W.D., Williams, C., de Olivera, P.E. & Krisel, C. (2007) Holocene fire and occupation in Amazonia: Records from two lake districts. Philosophical Transactions of the Royal Society of London (B) 362, 209-218. http://dx.doi.org/10.1098/rstb.2006.1980

Cordeiro, R.C., Turcq, B., Moreira, L.S., Rodrigues, R.d.A.R., Lamego Simões Filho, F.F., Martins, G.S., Santos, A.B., Barbosa, M., Guilles da Conceição, M.C., Rodrigues, R.d.C., Evangelista, H., Moreira-Turcq, P., Penido, Y.P., Sifeddine, A. & Seoane, J.C.S. (2014) Palaeofires in Amazon: Interplay between land use change and palaeoclimatic events. Palaeogeography, Palaeoclimatology, Palaeoecology 415, 137-151. http://dx.doi.org/10.1016/j.palaeo.2014.07.020

Hammond, D.S., Steege, H.t. & Van Der Borg, K. (2007) Upland Soil Charcoal in the Wet Tropical Forests of Central Guyana. Biotropica 39, 153-160. http://dx.doi.org/10.1111/j.1744-7429.2006.00257.x

---

## Referee Comment (RC2) · Anonymous Referee #2 · 8 Aug 2017

Le Page et al. made some (simple) improvements in the prognostic fire model HES-FIRE followed by parameter optimization. After the model is properly evaluated, they used it to predict future patterns of understory fires in Amazon forests under the CMIP5 RCP4.5 and RCP 8.5 scenarios. They found that land use change and climate change have a synergic role in strengthening fire activities in the RCP 8.5 scenario, with climate change exerting a dominant role, while conservative land use change under the RCP 4.5 scenario can actually mitigate fire occurrences. They also show that fire sizes will largely increase under both scenarios.

It is already known from previous studies that degradation fires (though not all of them are understory fires) in Amazon forests are largely controlled by drought conditions in relation to climate variations (Malhi et al., 2009, PNAS), and land fragmentation and

logging tend to increase the flammability of forests (Malhi et al., 2008 Science, Nepstad et al., 1999 Nature). Morton et al. (2008, GCB) shows that fire is an important agent in active management of agricultural lands after deforestation, for both pasture and croplands. In Morton et al. (2013) it is further shown understory fires are highly linked with deforestation frontiers (which is essentially land use change) and respond strongly to dry climate years or in general, to dry climate conditions. So in view of these studies, the conclusions in the current manuscript are not really very novel. But I recommend it being considered for publication for two reasons: (1) it incorporates the understory fires that are often neglected in global fire models. (2) it can provide useful insights for the future mitigation strategies for Amanzon forests.

Some general comments:

My general comments mainly concern with improving the presentation, especially to be more precise in the texts.

I find that the introduction section is written in a too much general and somewhat "loose" manner. For example, page 2, line 1-2 could be expanded to give more details. Descriptions in Page 2, line 23–26 is also too general, expressions like "predictable patterns of drought and fire risk form the basis of regional early-warming systems" could essentially apply on other fire types as well (e.g., boreal fires). The background of the current study is relatively well described, but I have a sense that it lacks a specific context that allow readers appreciating and better understanding the current study. For example, how about previous works by Alencar et al. 2004 (Ecological Applications) and Silvestrini et al. 2011 (Ecological Applications)? What are the progresses of the study in comparison to previous studies like these? The authors can also to think to enhance the specificity in the discussion section as well.

The flow of texts, to my point of view, sometimes lacks the necessary rigour needed in scientific writing. For example, page 2, line 24, "under a changing climate": although readers could guess from the contexts that the authors imply global warming or climate

change, or more specifically, climate change that induces more frequent drought, I still think it's better the authors directly write it out precisely as they intend to mean. Some other examples include: page 2 line 34, "under novel climate and land use conditions", what do you mean by "novel" here? Page 3 line 12, "... while addressing their respective issues... ", what are these respective issues? Page 4 line 10, "... MODIS patterns appeared more consistent with the contemporary distribution of land use ...", how such a conclusion is reached?

Minor comments: Page 4 line 15: in this equation, what are terms originally included in the HESFIRE in Le Page et al. (2015)? What are the new terms added accounting for understory fires? In section 3.1, could you explain how a better agreement between model and data is achieved? Is the inclusion of the extra term (land fragmentation impact on fire size) critical, or a recalibration of the original parameters more critical? (The authors could give some words based on their experts on their model, not necessarily with new simulations) I have a feeling like the interannual variability of the original model result is OK but just its magnitude is too high (Figure S2), so that an extra term is needed to bring down the burned area. Visually looking Figure 2(c) is quite OK but could you show a scatter plot (model versus observation) as well (maybe in the supplement)? Finally, how the land fragmentation is measured in the model? Like you used some land cover map derived index?

Figure 3 and the associated results: Are these percentiles calculated by pooling on over each grid cell the results from different climate models? Is there a risk that the fires could be overestimated because different climate models give different spatial patterns of drying (Fig 1 B)? I mean, spatially we pick up the 90th percentile over each pixel so that the spatial total of the 90th percentile fires are much larger than, if we just pick up the 90th of total fire impacted areas from different models, because models compensate for each other spatially?

---

## Author Comment (AC1) · 9 Oct 2017

**Reviewer #1**

As a specialist in past ecological change, and having worked with reconstructing fire histories in the Amazon basin, I was interested to read the discussion manuscript by Le Page et al. related to the likely impact of future climate change on fire in Amazonia. In this manuscript Le Page et al. develop the global intermediate complexity Human-Earth System FIRE (HESFIRE) model (Le Page et al. 2015) to explore fire dynamics in the Amazon basin. The new outputs presented here project how understory fire activity could change within Amazonia over the next c. 100 years under two different future climate change scenarios (Representative Concentration Pathways 4.5 and 8.5;Thompson et al. 2011, Riahi et al. 2011). The HESFIRE model is evaluated favourably against observed fire activity between 1999 and 2010; interestingly the new parameterisation of the model (this manuscript) is shown to be a considerable improvement on the original parameterisation used in the global model (compare Figure 2 with FigureS2). The authors predict that over the coming decades climatic change will lead to changes in relative humidity, temperature and fuel moisture that will lead to increased duration and size of fires, and that fires will impact a larger area of the basin.

The manuscript presented by Le Page et al. is clearly written, well organised and contains clear figures (notwithstanding minor comments below). The model parameters make intuitive sense to me but I am not an expert in this area. The conclusion that the synergy of land-use (humans) and climate change are likely to increasing fire risk in Amazon basin is reasonable based on the data presented. It also does not come as a surprise to me as there is a long history of human-climate-fire interactions within the Amazon basin. Fire (most probably triggered by early human populations) has been a component of Amazonia for c. 8,000-9,000 years (e.g. Bush et al. 2007; Hammond et al. 2007), and the impact of this fire has likely been modulated by past climate change (Bush et al. 2008; Bush et al. 2017; Cordeiro et al. 2014). Indeed Bush et al. (2017) make a specific link to past changes in the El Nino Southern Oscillation(ENSO), human activity, and changes in the Amazonian fire regime (based on fossil charcoal records). Given that the message from the past, and the projections for the future, seem to be providing consistent message (humans and climate synergistically modifying fire regimes within Amazonia) it might be interesting to add a short section of discussion to this manuscript to make this comparison. This could fit well at the start of the discussion section where ENSO is discussed in the context of future predictions.

We appreciate the reviewer's suggestion to better integrate our contemporary and future examination of fire dynamics in the Amazon with insights from paleo studies. We highlighted the consistency with paleo studies in introduction and discussion:
In introduction (P.2 l.25): "Satellite observations over the last 40 years have confirmed the importance of these fire-climate-land use interactions (Alencar et al., 2015; Chen et al., 2013; Laurance, 1998), consistent with paleorecords of charcoal accumulation rates inferred from sedimentary cores (Bush et al., 2007; Cordeiro et al., 2014)."

In addition, in the discussion we note that (P.8 l.20): "Model results with RCP4.5 land use projections suggest that a significant and regional-scale reduction in agricultural activities and landscape fragmentation can disrupt these climate-fire interactions. This is consistent with contemporary studies (Alencar et al., 2004; Morton et al., 2013), and with the clear signature of land use activities in charcoal paleorecords since the onset of Amazonian agriculture (Bush et al., 2007; Cordeiro et al., 2014). In fact, (Bush et al., 2017) analyzed a 6900 years sedimentary record in western Amazonia, and found that the

ENSO-fire signal was strongly expressed at times of widespread agricultural activity, being otherwise undetectable in the record and most likely absorbed by natural vegetation."

**MINOR COMMENTS**
**Page 3, line 10-20: There seems to be a bit of a jump in the HESFIRE model. It would be useful to add some linking text that makes clear what this model is, how it has been previously used and why the work presented her is a good next step.**

In the revised manuscript, we have expanded the text to discuss previous applications of the model at a global scale and clarified the basis for applying the model in regional studies such as this one:

P.3 l.20: "The model has been applied at global scale (Le Page et al., 2015) and used in a sensitivity experiment to evaluate the propagation of uncertainties from land cover and climate input data to estimates of fire activity (Le Page, 2016). The HESFIRE model was designed to facilitate the development of regional versions – a capability used in leveraged in this Amazon-scale study - with the integration of a data assimilation component to regionally adjust the parameterization of fire drivers based on observed fire dynamics."

**Page 4, line 13: Additional space between "Figure" (also: page 6 line 11, page 7 line30, page 8 line 29, page 10 line 10).**

Thanks !

**Page 8, line 33: Referencing anonymous new data seems a bit odd. Can you find a clearer way of referencing the source of these data?**

We have corrected the citation format for these data. The correct reference is a website with near real-time estimates of fire risk in the Amazon based on Chen et al., 2011 (also cited). We now provide the URL directly (https://www.ess.uci.edu/~amazonfirerisk/).

**Figures: Various fonts are used on the different figures. I think it would help the manuscript to look more coherent if these were standardised.**
We agree that more uniform fonts would improve the figures. If the paper is accepted for final publication, we will harmonize fonts to the extent possible.

---

## Author Comment (AC2) · 9 Oct 2017

**Reviewer #2**

Le Page et al. made some (simple) improvements in the prognostic fire model HESFIRE followed by parameter optimization. After the model is properly evaluated, they used it to predict future patterns of understory fires in Amazon forests under the CMIP5 RCP4.5 and RCP 8.5 scenarios. They found that land use change and climate change have a synergic role in strengthening fire activities in the RCP 8.5 scenario, with climate change exerting a dominant role, while conservative land use change under the RCP 4.5 scenario can actually mitigate fire occurrences. They also show that fire sizes will largely increase under both scenarios. It is already known from previous studies that degradation fires (though not all of them are understory fires) in Amazon forests are largely controlled by drought conditions in relation to climate variations (Malhi et al., 2009, PNAS), and land fragmentation and logging tend to increase the flammability of forests (Malhi et al., 2008 Science, Nepstad et al., 1999 Nature). Morton et al. (2008, GCB) shows that fire is an important agent in active management of agricultural lands after deforestation, for both pasture and croplands. In Morton et al. (2013) it is further shown understory fires are highly linked with deforestation frontiers (which is essentially land use change) and respond strongly to dry climate years or in general, to dry climate conditions. So in view of these studies, the conclusions in the current manuscript are not really very novel. But I recommend it being considered for publication for two reasons: (1) it incorporates the understory fires that are often neglected in global fire models. (2) it can provide useful insights for the future mitigation strategies for Amazon forests.

Some general comments:
My general comments mainly concern with improving the presentation, especially to be more precise in the texts. I find that the introduction section is written in a too much general and somewhat "loose" manner. For example, page 2, line 1-2 could be expanded to give more details. Descriptions in Page 2, line 23–26 is also too general, expressions like "predictable patterns of drought and fire risk form the basis of regional early-warming systems" could essentially apply on other fire types as well (e.g., boreal fires).
The flow of texts, to my point of view, sometimes lacks the necessary rigour needed in scientific writing. For example, page 2, line 24, "under a changing climate": although readers could guess from the contexts that the authors imply global warming or climate change, or more specifically, climate change that induces more frequent drought, I still think it's better the authors directly write it out precisely as they intend to mean. Some other examples include: page 2 line 34, "under novel climate and land use conditions", what do you mean by "novel" here?

We appreciate the suggestions to clarify the text. We have carefully edited the manuscript to avoid confusion with both general concepts and specific references to previous research on fire activity in Amazonia.

**Page 3 line 12, "...while addressing their respective issues...", what are these respective issues?**

The specific issues include vegetation error propagation in DGVM-fire models, and the 1-day limit imposed for fire duration, while multi-day fires are an essential aspect of understory fires. This has been clarified in the text:

P.xx, l.xx: "HESFIRE is a fire model of intermediate complexity seeking to combine the explicit fire representation in dynamic global vegetation models (DGVM-fire models)

with the performance of statistical fire models, while addressing some of their issues (Le Page et al., 2015). In particular, land cover distribution in HESFIRE is inferred from contemporary observations, avoiding error propagation from the vegetation scheme to the fire module, which is a recurrent challenge in DGVM-fire models (Kelley et al., 2013; Kelley and Harrison, 2014; Wu et al., 2015). HESFIRE was also designed to represent multi-day fires, tracking each individual fire on a 12-hour time steps, whereas other global fire models have a maximum fire duration of 1 day (Arora and Boer, 2005; Li et al., 2012; Thonicke et al., 2010). The model has been applied at global scale (Le Page et al., 2015) and used in a sensitivity experiment to evaluate the propagation of uncertainties from land cover and climate input data to estimates of fire activity (Le Page, 2016). The HESFIRE model was designed to facilitate the development of regional versions – a capability used in leveraged in this Amazon-scale study - with the integration of a data assimilation component to regionally adjust the parameterization of fire drivers based on observed fire dynamics."

**Page 4 line 10, "...MODIS patterns appeared more consistent with the contemporary distribution of land use...", how such a conclusion is reached?**

There are large regional-scale discrepancies in land use density between the MODIS and GLOBcover products (see figure 18.2a,c in (Le Page, 2016)). We chose MODIS based on expert knowledge in the team, and on a visual comparison of both datasets to other sources of information/knowledge on the regional distribution of land cover, especially agriculture (e.g. (Soares-Filho et al., 2014)). We now mention in the text:

P.xx, l.xx: "Although there is no comparison study of both datasets in the Amazon, MODIS patterns appear more consistent with the contemporary distribution of land use, as inferred from expert knowledge in the team and from a comparison with other sources of information (e.g. (Soares-Filho et al., 2014))."

**The background of the current study is relatively well described, but I have a sense that it lacks a specific context that allow readers appreciating and better understanding the current study. For example, how about previous works by Alencar et al. 2004 (Ecological Applications) and Silvestrini et al. 2011 (Ecological Applications)? What are the progresses of the study in comparison to previous studies like these? The authors can also think to enhance the specificity in the discussion section as well.**

We clarified in the introduction the novelty of the study for understory fires in the Amazon (P.2 l.27): "Projections of Amazon fire activity also suggest strong synergies between climate change and anthropogenic expansion scenarios (Cardoso et al., 2003; Le Page et al., 2010; Silvestrini et al., 2011), but previous work focuses primarily on deforestation and agricultural burning. **These fires are managed, burn different types of fuels, and are generally of short duration, thus provide few insights about the ecology of slow-moving, multi-day understory fires.** Understory fires are difficult to detect using satellite data because they do not burn the forest canopy, and only a few studies have inferred their extent in small regions to explore their dynamics and drivers (Alencar et al., 2006, 2004; Ray et al., 2005). However, a method was recently developed to detect understory forest fires using multi-year satellite image time series (Morton et al., 2011, 2013). These Amazon-wide observations provide a critical foundation to develop simulations of understory fire dynamics under novel climate and land use scenarios."

**Minor comments:**
**Page 4 line 15: in this equation, what are terms originally included in the HESFIRE in Le Page et al. (2015)? What are the new terms added accounting for understory fires?**

The whole equation is new, as fire ellipses were fully burned in the original version of the model (BA = E). We have now clarified this in the text:

P.4 l.18: "A new equation was developed in this study to compute the area that actually burns as a fraction of the plain ellipse, driven by landscape fragmentation and fire weather:"

$$BA = E \times \left(1 - F_n^{Fexp}\right) \times \left(1 - RH_n^{RHexp}\right) \times \left(1 - SW_n^{SWexp}\right) \times \left(1 - T_n^{Texp}\right)$$

**In section 3.1, could you explain how a better agreement between model and data is achieved? Is the inclusion of the extra term (land fragmentation impact on fire size) critical, or a recalibration of the original parameters more critical?(The authors could give some words based on their experts on their model, not necessarily with new simulations). I have a feeling like the interannual variability of the original model result is OK but just its magnitude is too high (Figure S2), so that an extra term is needed to bring down the burned area.**

As clarified in the previous section, equation (2) was entirely new in the model: fire size was affected not only by land fragmentation, but also by fire weather.
We did runs with 3 configurations of the model to explore the impact of the new fire size equation and of the regional parameterization (see Figure R1 below). When only the fire size equation is appended to the original model, burned areas are clearly lower and more realistic, and inter-annual variability is also improved (e.g. the 2005 peak is reduced). When the parameters were re-optimized, both burned areas (e.g. 1999, 2001-2002) and inter-annual variability were also affected, although not as much.

We now cite this discussion in the paper: "The regional version of HESFIRE reproduced the observed spatial patterns of average fire activity, including the clear boundary between fire-affected forests along the arc of deforestation and mostly fire-free forests in more humid regions of the central and western Amazon with less agricultural activity (Figure 2a, Figure S2, see also reply to referee comment #2 about the contribution of model adjustments to improved performances*)."

* All interactive comments in ESDD are fully citable. Citation format to be modified according to the journal standards.

[Figure]

**Figure R1: 1999-2010 annual burned areas under 3 HESFIRE configurations.**

**Visually looking Figure 2(c) is quite OK but could you show a scatter plot (model versus observation) as well (maybe in the supplement)?**

We have generated an alternate version of Figure 2 (panels b and c).

[Figure]

We added the figure in supplementary material if readers want to get a closer look on this aspect. We maintained the short discussion in the paper unchanged ("fire size distribution consistent with observations").

**Finally, how the land fragmentation is measured in the model? Like you used some land cover map derived index?**
Yes, the fragmentation in a grid-cell index is calculated as the fraction of land covers that represent a significant barrier to fire propagation. We have now clarified this information in the model overview section:

(P.4, l.5): "- *Fire termination.* Four factors control the termination of fires: a) a change to non fire-prone weather conditions (e.g. fires terminate when relative humidity increases above 80%); b) low fuel availability (the probability of termination is higher in sparsely-vegetated landscapes); c) landscape fragmentation (**the fraction of a grid-cell covered by croplands, urban areas, water bodies, bare areas, and burned areas over the last 8 months);** and d) fire suppression efforts, which intensify with higher land use density and GDP, but become less efficient under increasingly fire-prone weather."

**Figure 3 and the associated results: Are these percentiles calculated by pooling on over each grid cell the results from different climate models? Is there a risk that the fires could be overestimated because different climate models give different spatial patterns of drying (Fig 1 B)? I mean, spatially we pick up the 90th percentile over each pixel so that the spatial total of the 90th percentile fires are much larger than, if we just pick up the 90th of total fire impacted areas from different models, because models compensate for each other spatially?**

The reviewer is correct in that Figure 3 shows the gridcell-level percentiles among different climate models. As such, none of them is a regional-scale output of one given model/scenario. We feel it gives a better representation of the ensemble runs. Picking one model run that results in a 90th percentile of region cumulative burned areas, for example, would focus the results/discussion on the area most heavily impacted by that given climate model projections. Other areas would be seemingly resilient to climate change, but it would appear this way only due to that specific model.

Given that we also show the 10th percentile and median, we believe this gives an adequate view of fire projections among the ensemble runs. Regional-scale burned areas for each

model/scenarios are shown in Figure S5. We have clarified the underlying method in the caption of figure 3:

New caption: "Figure 3: Annual burned fraction projected in HESFIRE for 2080-2100 varies across models and climate scenarios. The 10th, 50th and 90th percentiles are **calculated at the grid-cell level among the 8 climate runs** (see Methods)."

---

## Author Response (AR2)

**Editor comments:**

**I finally had a chance to read carefully your manuscript, and it appears clearly that you have taken great care of the writing of the manuscript and creation of figures. The manuscript reads very well, sentences are carefully constructed and easy to read even for a non-specialist of fires, as me. I have to say no comment, except perhaps one or two technicalities and a few clarifications required on scientific aspects:**

**- ESD will appreciate dois of references, both in the main manuscript and in the supplementary material.**
**- I am unsure of the "n.d." convention for papers which do not have a date, yet, but the publishing office will take care of that anyway.**
**- page 8 line 21 : extra parentheses around Bush 2001**

The references are generated automatically by Zotero, and don't always come with DOIs indeed. To change this I'll need to remove zotero automatic fields and edit by hand. If it is ok I will keep them this way for now in case zotero is needed for some changes (including on your comments below), and will add the required information once everything else is settled.

**- Equation (1) is the one that triggered some scientific comments of mine. As you say that you use a Monte-Carlo technique, you must take care of the stochastic variance of the parameters. In which case you must be using a kind of Metropolis-Hastings algorithm, which actually requires a _likelihood_ function (not just a cost function). Can you clarify ?**
Statistics are not really my area of expertise, but one of the co-authors who understands them better suggested this simple edit to the manuscript:
P. 4, l. 29: "The MCMC likelihood function employs an optimization metric, which combines average burned area and inter-annual variability at the grid-cell level".

Does that address your comment ?

**- Figure 3 : "indicate variability from 8 climate models" : I have some difficulties here. Isnt't it rather "uncertainty" (using the inter-model spread as a measure of uncertainty, which is standard practice tough not free of questions), and in that case what is the relevance of speaking in terms of percentiles when there are only 8 samples? Please clarify.**
We used the term "variability" instead of "uncertainty" because climate models have little (or no) predictive value. I don't feel comfortable talking about uncertainty, while the reality in 50 years could very well be beyond the range of projections from these models. So the percentiles show the spread within this group of model, but do not quantify uncertainties about future climate change. Does that make sense ? We mentioned low consensus/predictive value of climate projections in the first discussion paragraph, but shall I further edit the manuscript to clarify ?

**Non-public comments to the Author:**
**I recommended your paper for a 'highlight' in EGU.**
Thank you very much ! And many thanks for your support and editorial work !